# Ethnic Issues and Disparities in Inflammatory Bowel Diseases: What Can We Learn from the Arab Population in Israel?

**DOI:** 10.3390/jpm13061008

**Published:** 2023-06-17

**Authors:** Naim Abu-Freha, Nour Ealiwa, Muhammad AbuTailakh, Muhammad Abu-Abed, Sarah Bader, Rachel Tabu, Doron Schwartz

**Affiliations:** 1The Institute of Gastroenterology and Hepatology, Soroka University Medical Center, Beer-Sheva 84101, Israeldoronsh@clalit.org.il (D.S.); 2Faculty of Health Sciences, Ben Gurion University of the Negev, Beer-Sheva 84105, Israel; 3Department of Nursing, Faculty of Health Sciences, Recanati School for Community Health Professions, Ben Gurion University of the Negev, Beer-Sheva 84105, Israel; 4Nursing Research Unit, Soroka University Medical Center, Beer-Sheva 84101, Israel; 5Internal Medicine Division, Soroka University Medical Center, Beer-Sheva 84101, Israel

**Keywords:** Crohn’s disease, ulcerative colitis, Israel, Arab, inflammatory bowel disease

## Abstract

Inflammatory bowel diseases are increasing among different ethnic groups. We aimed to compare the clinical characteristics, complications, and outcomes among Arab and Jewish people sharing the same healthcare system. All patients older than 18 years with a diagnosis of Crohn’s disease (CD) or ulcerative colitis (UC) between the years 2000 and 2021 were included. Data regarding demographics, disease characteristics, extraintestinal manifestation, treatment, comorbidities, and mortality were retrieved. A total of 1263 (9.8%) Arab CD patients were compared with 11,625 Jewish CD patients, and 1461 (11.8%) Arab UC patients were compared to 10,920 Jewish patients. Arab CD patients were younger at diagnosis, 36.11 ± 16.7 compared to 39.98 ± 19.4 years, *p* < 0.001, 59.5% males compared to 48.7%, *p* < 0.001; in addition, Arab CD patients had a higher rate of anal fissure, perianal abscess, erythema nodosum, diabetes mellitus, obesity, liver cirrhosis, and male infertility. Arab CD patients were less frequently treated with azathioprine or mercaptopurine compared with Jewish patients. No significant difference was found in the rate of anti-TNF treatment, but a higher rate of steroids treatment was found. The all-cause mortality of CD patients was lower among Arab patients (8.4% vs. 10.2%, *p* = 0.039). Significant differences were found regarding disease characteristics, course, comorbidities, and treatment among Arab and Jewish patients with IBD.

## 1. Introduction

Race and ethnicity play important roles in the epidemiology and outcomes of diseases; this issue has been investigated in different diseases in the field of gastroenterology. Specifically, the incidence, prevalence, serologic, and genetic markers of inflammatory bowel disease (IBD) vary significantly by population, race, and ethnicity [1,2,3,4,5,6,7,8,9]. A previous study including 31 European centers found a significant gradient in the incidence of IBD between the East and West [1]. In addition, significant racial and ethnic differences of IBD were found over four decades (between 1970 and 2010) in the United States population, with an increase in the incidence rate by 39% among whites and 139% among nonwhites [10]. Another study found a high age-adjusted difference in incidence of IBD among Indians compared to white Europeans and Pakistanis in the United Kingdom [11]. While the relationship between IBD prevalence and race and ethnicity has been previously studied [2,5,6,10,11], few studies have investigated the clinical course and outcomes of IBD from an ethnic perspective.

Studies have shown an increase in the incidence and prevalence of IBD among the Arab population in Israel [3,4,5,6,7]. However, disparities regarding other aspects such as NOD2/CORD15 and serological markers (ASCA/ANCA) were found among different ethnic groups in Israel and other countries [7,8,9,12,13]. This Arab population in Israel is of specific interest due to its unique historical, social, and cultural characteristics. Arabs in Israel have undergone significant and rapid changes in lifestyle over the past five decades, including shifts from agricultural lifestyles to urban lifestyles, followed by changes in diet, increases in rates of smoking among men, decreases in rates of physical activity, and higher rates of consanguinity and genetic diseases. All of these factors impact the health of the population [14,15], particularly environmentally influenced diseases such as IBD. 

Notably, studies published up until this point have included a small number of patients in Israel and small percentages of ethnic groups in general. A systemic review of IBD in the Arab world was published recently, showing an increase in the IBD incidence, a pooled incidence rate of 2.33 (95% CI 1.2–3.4) per 100,000 persons per year for UC, and a pooled incidence rate of 1.46 (95% CI 1.03–1.89) per 100,000 persons per year for CD [6].

Understanding the effects of race and ethnicity on IBD provides insight into the clinical course, disease behavior, coping mechanisms, and disease outcomes among different patients, providing a new path for diagnostic and therapeutic improvements. In this national-based study, we aimed to investigate IBD among the Arab population in Israel regarding epidemiology, comorbidities, extraintestinal manifestations, complications, treatments, and all-cause mortality in comparison to the Jewish population.

## 2. Materials and Methods

### 2.1. Patients 

This is a retrospective, population-based, observational study. We enrolled all adult patients (age 18 years and older) with a confirmed inflammatory bowel disease diagnosis, who were identified as described before [16]. Only patients who had a diagnosis of CD or UC according to the ICD-9 codes and were treated at least for three months with IBD treatment were considered to have IBD and included in the study. The inclusion criteria specified adults with CD or UC diagnosis and treated for the disease; patients with unclassified IBD or never treated for IBD were excluded. The study included data between the years 2000 and 2021, extracted from Clalit Health Services (CHS) database, a health maintenance organization in Israel, using a platform powered by MdClone (https://www.mdclone.com, accessed on 10 May 2023). CHS is one of the largest health maintenance organizations worldwide, with about five million insured residents, representing over 53% of the Israeli population and over 75% of the Arab population in Israel. The MdClone platform includes computerized data of the patients, data from the community clinics, emergency department, and hospitals. Data are available for every patient, including diagnoses, events such as hospitalizations, emergency department visits, surgeries, laboratory results, and medication prescriptions. Any new diagnosis, hospitalization, or medical updates is recorded as a new event. Exploring events on a linear timeline is possible due to the longitudinal data organization. Anonymous patient data can be extracted in relation to other recorded events. The MdClone platform enables exploration of large sizes of cohorts, long-term data collection, and a holistic view of the patients from birth to death.

### 2.2. Data Collection 

Data regarding IBD diagnosis, demographics, extraintestinal manifestations, treatment, surgery, and complications among Arabs and Jews were collected. In addition, data regarding common laboratory values (hemoglobin, white blood count, platelets, liver enzymes, albumin, C-reactive protein, and calprotectin), nearest to the time of IBD diagnosis within six months before or after diagnosis, were recorded. The lab values for every variable were included in the tables if data were available among at least 80% of the patients, except calprotectin, which was not available in the entire study period. Data regarding all-cause mortality was also included. The patients and results were subdivided into those with a diagnosis of Crohn’s disease (CD) and those with a diagnosis of ulcerative colitis (UC). Arab and Jewish patients were also analyzed separately for comparison using a predefined variable from the platform that is based on clinic and village/city. 

The study was carried out in accordance with the principles of the Helsinki Declaration. The study protocol was approved by the Institutional Helsinki Committee, approval number 97–21. Informed consent was waived due to the retrospective design of the study.

### 2.3. Statistical Analysis 

Data are presented as mean ± SD for continuous variables and as a percentage (%) of the total for categorical variables. Univariate analyses were performed by the Mann-Whitney test (for continuous variables), Fisher’s exact, and chi-square tests (for categorical variables). All statistical analyses were performed using IBM SPSS version 26 (Chicago, IL, USA). Any *p*-values less than 0.05 were considered statistically significant. 

We used logistic regression models to examine the multivariate relationships between risk factors and mortality and hospitalization among CD and UC patients. Before introducing the variables into the model, the multicollinearity of the variables was examined using the Variance Inflation Factor (VIF) statistic. 

## 3. Results

A total of 25,269 IBD patients were included in our analysis: 12,888 (51%) CD patients and 12,381 (49%) UC patients. A total of 1263 (9.8%) Arab CD patients were compared to 11,625 Jewish CD patients, and 1461 (11.8%) Arab UC patients were compared to 10,920 Jewish patients. 

### 3.1. Patients Proportion for Two Decades 

Our data showed a continuous increase in the proportion of Arab IBD patients among all patients diagnosed in the same year, for both CD and UC, as seen in Figure 1 and Figure 2.

### 3.2. Crohn’s Disease 

The average age at diagnosis among Arab patients was 36.11 ± 16.7 compared to 39.98 ± 19.4 years in Jewish CD patients, *p* < 0.001, with a higher percentage of patients in the younger age groups and a lower percentage in the group aged 65 and older. A total of 59.5% of Arab IBD patients were males compared to 48.7% of Jewish IBD patients, *p* < 0.001. A higher rate of smoking was also found among Arab patients (36.1% vs. 31.2%, respectively, *p* < 0.001). The baseline characteristics of the two populations are summarized in Table 1. Regarding extraintestinal manifestations, Arab CD patients had higher rates of anal fissure, peri-anal abscess, and erythema nodosum (6.2 vs. 4.9, *p* = 0.053, 9.7% vs. 7.3% *p* = 0.003, 2.4% vs. 1.4% *p* = 0.035, respectively,) and a significantly lower rate of uveitis (1.8% vs. 5.6%, *p* < 0.001).

Significant differences were found between Arab and Jewish CD patients in terms of comorbidities (Appendix A); significantly higher rates of diabetes mellitus, obesity, liver cirrhosis, and male infertility was found among Arab CD patients (12.2% vs. 8.6%, 28.3% vs. 21.6%, 1.7% vs. 1%, 6.6% vs. 2.5%, *p* < 0.001, respectively), while significantly lower rates of hypertension, fatty liver, dementia, and depression were found among these patients (23.8% vs. 27.8%, 6.3% vs. 8.5%, 1.3% vs. 3%, 12.7% vs. 19.5%, *p* = 0.003, 0.006, 0.001, and <0.001, respectively). In addition, lower rates of melanoma and basal cell carcinoma was observed among Arab CD patients (0.2% vs. 1.4% and 2% vs. 10.8%, *p* < 0.001, respectively). There were no significant differences in the rates of ischemic heart disease, chronic lung disease, dyslipidemia, iron deficiency anemia, and colon cancer. 

The rate of hospitalization was higher among Arab CD patients, with 85.4% of the patients hospitalized at least one time, compared to 78.4% of Jewish CD patients, (*p* < 0.001). Additionally, the mean number of hospital admissions was 5.96 ± 8.4 times among Arab CD patients compared to 4.7 ± 6.9 times among Jewish CD patients (*p* < 0.001).

We compared the treatment profile (Table 2) and found that Arab CD patients were less frequently treated with azathioprine and mercaptopurine compared with Jewish patients. No significant difference was found in the rate of anti-TNF use (38.2% vs. 40.4%, *p* = 0.121), but the time from CD diagnosis to anti-TNF treatment was significantly shorter among Arab compared to Jewish patients, 1121 ± 1334 vs. 1409 ± 1484 days, *p* < 0.001. A higher rate of steroid (prednisone) treatment was found among Arab CD patients (76.6% vs. 64.1%, *p* < 0.001). Lower rates of vedolizumab and ustekinumab were also found among Arab CD patients (7.2% vs. 10.3% and 6 vs. 8.3%, respectively). No significant difference was found regarding bowel surgery rate among Arab and Jewish patients. The all-cause mortality of CD patients was lower among the Arab patients (8.4% compared to 10.2%, *p* = 0.039), but the age at death was younger (64 ± 18.4 years compared to 74.7 ± 15.2 years, *p* < 0.001). 

Focusing on laboratories values at the diagnosis time, higher levels of C-reactive protein (CRP) (10.34 ± 31.1 vs. 5.67 ± 18.3, *p* < 0.001), white blood cells count (8.99 ± 3.4 vs. 8.53 ± 3.3, *p* < 0.001), and platelets (326.2 ± 121 vs. 316.56 ± 11.6, *p* = 0.007) were found among Arab CD patients. In contrast, lower iron was found (50.73 ± 34 vs. 60.58 ± 38, *p* < 0.001); results are summarized in Table 3. 

### 3.3. Ulcerative Colitis 

A total of 1461 (11.8%) Arab UC patients were compared to 10,920 Jewish UC patients, with a mean age of diagnosis of 37.9 ± 16.2 years vs. 45.6 ± 19.5 years, respectively (*p* < 0.001). A higher percentage of Arabs were male (52.6% vs. 48.1%, respectively, *p* = 0.001). The baseline characteristics of these patients are summarized in Table 4.

Arab UC patients had a lower rate of uveitis and arthritis but no significant difference regarding other extraintestinal manifestations (Table 4). 

Appendix A summarizes the comorbidities of the study population. Significantly lower rates of ischemic heart disease, hypertension, dyslipidemia, chronic renal failure, dementia, and depression were found among Arab UC patients compared to Jewish patients (13.9% vs. 18.5%, 24.6% vs. 35%, 32.2% vs. 42.6%, 7.3% vs. 11.5%, 1.9% vs. 4.1%, and 1.8% vs. 19.3%, *p* < 0.001, respectively,) while a significantly higher rate of obesity was observed among Arab patients, (29% vs. 23%, *p* < 0.001). A lower rate of colorectal cancer and skin cancers (melanoma and basal cell carcinoma) were found among Arab UC patients. 

Arab UC patients were treated more frequently with azathioprine (17.7% vs. 10.1%, *p* < 0.001) and anti-TNF (17.9% vs. 13.6%, *p* < 0.001). A higher but nonsignificant treatment rate of vedolizumab (10.4% vs. 9%, *p* = 0.071) and a higher rate of steroids (prednisone) were prescribed (70.5% vs. 55.1%, *p* < 0.001) than Jewish patients. Treatment of anti-TNF and vedolizumab was at a younger age among Arab UC patients and at a shorter time from diagnosis. No significant difference was found regarding the rate of bowel surgery. Results regarding treatment are summarized in Table 5. 

Laboratories at the diagnosis time are summarized in Table 6; recorded are higher levels of CRP, hemoglobin, WBC, and platelets among Arab UC patients compared to Jewish UC patients, (5.75 ± 19.2 vs. 4 ± 17.5, 12.87 ± 2 vs. 13.1 ± 1.75, 299.1 ± 107 vs. 282.8 ± 96, *p* < 0.001, respectively).

### 3.4. Univariate and Multivariate Analysis 

In the multivariate model, we found a relationship between the ethnicity and the mortality and hospitalization among CD and UC patients. Arab IBD patients have a lower risk for all cause mortality compared to Jewish IBD patients (Crohn’s disease patients; OR 0.810, *p* = 0.047; 95% CI [0.657–0.997], Ulcerative colitis patients; OR 0.544, *p* < 0.001; 95% CI [0.448–0.661]). In addition, a lower risk for hospitalization was found among Arab IBD patients (Crohn’s disease patients (OR 0.524, *p* < 0.001; 95% CI [0.421–0.652], Ulcerative colitis patients (OR 0.429, *p* < 0.001; 95% CI [0.332–0.554]). The results of the univariate and multivariate models are presented in Table 7. 

## 4. Discussion

Clinical characteristics, disease course, and outcomes of IBD may be influenced by ethnicity. Our study focused on the Arab population in Israel. It included IBD patients of two different ethnicities sharing the same healthcare system and extracted from a big data platform. We found significant differences in terms of disease characteristics, treatment, complications, and comorbidities. 

There is a growing incidence of IBD in the Arab world, and IBD patients from Arab countries may present some different characteristics compared to their European counterparts. In this study, we showed a continuous increase in Arab IBD patients, with a proportion reaching more than 15% of all patients diagnosed annually. As published previously, there is an increase in IBD not only among the Arab population in different countries but globally [2,5,6,17]. The previous report showed that the number of Jewish IBD patients doubled in Israel between the years 2005 and 2018, while the number of Arab patients increased threefold during the same period [18]; in this study, the prevalence of IBD was increasing but still gradually decelerating [18]. Despite the global increase in IBD prevalence and incidence, it seems that these increased rates are higher among specific populations, such as the Arabs in Israel. In general, the three most important factors that increase the risk of IBD development are: genetic susceptibility, environmental factors, and microbiomes. Increases in IBD have been observed specifically in newly industrialized societies, which could be explained in part by lifestyle changes and the westernization of diet and environments, both of which influence the bowel microbiome and increase the risk for IBD among genetically susceptible people [12]. We reported a high rate of CD patients (32%) as carriers of the NOD2/CARD15 mutation among a subgroup of Arabs in Israel [7]. Only scant data are available regarding the genetics among IBD patients in the Arab population with a diverse mutation rate [9,19]. The Arab world in general, and particularly Arabs in Israel, have undergone enormous environmental changes over the last decades: major urbanization and Westernization of nutritional patterns, smoking rates, and antibiotic use in childhood have led to consequent changes in the intestinal microbiome, and all these together could be the cause for increase of IBD prevalence. However, in our study, the percentage of the Arab CD patients increased continuously in the last two decades, while a prominent decline was found in UC in the last two years (2020 and 2021). It is possible that this is because the COVID-19 pandemic led to a delay in UC diagnosis, particularly among patients with mild or moderate disease severity. Follow-up over the next years will help us to understand if this is a trend and will continue to decrease, or if it will be only a momentary decline with a reflex increase in the next years. 

In our large cohort of patients, we found several disparities between Arabs and Jews. Arab IBD patients were more likely to be male, diagnosed at a younger age, and fall into a younger age group than their Jewish counterparts. These characteristics can be explained by the young age of the Arab population in Israel; only about 5% of the Arab population in Israel is over the age of 65 years [20]. More Arab CD and UC patients were males than Jewish patients. Another study investigated the epidemiology of IBD among Arab patients in Israel and reported a higher rate of males among CD patients but not among UC patients [2]. Data from Western and Eastern Europe found a higher rate of males in newly diagnosed UC patients, but a male’s predominance was found among CD patients only in Eastern Europe [1]. In the meta-analysis of studies from the Arab World a male predominance reported, with 41–60% of the IBD patients [6] and other studies showed different findings according to the ethnic groups [10,11]. The higher frequency rate of males may be explained by different factors, such as exposure rates to specific risk factors like smoking or specific foods; however, this remains to be determined in future studies. Further, significant differences were found regarding the extraintestinal manifestations between the different ethnic groups. A higher rate of erythema nodosum and a lower rate of uveitis and arthritis were found among Arab UC patients compared to the Jewish IBD patients. 

In our study, we observed several findings among Arab IBD patients that can indicate a more severe disease course. We found a higher rate of hospitalized patients and a larger number of total hospitalizations among Arabs. In addition, CD patients had a higher rate of abdominal abscesses, a higher rate of steroid treatment, and a shorter time to biologic treatment. 

Furthermore, significant differences were found in treatment; a larger proportion of Arab patients with IBD were treated with azathioprine than Jewish patients. This could be related to patients’ preferences to be treated with oral treatment rather than intravenous or subcutaneous medications. Additionally, it is possible that other factors, such as the cost of biologics, resulted in the delay of biologics treatment. A higher rate of anti-TNF treatment was found among Arab patients with UC compared to Jewish patients, but there was no significant difference among CD patients. In all cases, the rate of anti-TNF treatment was much higher among Arabs in our study compared to the Arab population in other countries. In our study, 38.2% of CD patients and 17.9% of UC patients were treated with anti-TNF compared to 4–8.3% of patients in other Arab populations [6,21,22,23]. Furthermore, Arab IBD patients were treated with steroids (prednisone) more commonly than Jews in both diseases, reaching about 75% of the patients. 

The all-cause mortality of CD patients was lower among the Arab patients, 8.4%, compared to 10.2% of Jewish patients. However, the age at death was significantly younger, likely due to the difference in age and comorbidities among Jewish patients. 

In summary, significant differences regarding disease characteristics, clinical course, extraintestinal manifestation, comorbidities, complications, and treatment were found among different ethnic groups, Arabs and Jews. Disparities were found regarding age, gender predominance, hospitalization, extraintestinal manifestation, comorbidities, and death. These differences can be explained by different factors related to the patients (genetics, environmental factors, adherence to diagnostic procedures and treatment, medical awareness, healthcare seeking), healthcare accessibility and availability, and treatments. Different NOD2/CARD15 carrier rates were found among different ethnic groups, and different carrier rates were also found among subgroups of the Jewish population. A NOD2/CARD15 carrier rate of 32% among CD Bedouin patients was reported, 47.4% among Ashkenazi Jewish CD patients, 27.4% among Sephardic Jews [7,12], and 7–16% among Caucasian CD patients [13]. A lower carrier rate of 8.2% was found among Arabs in other regions in Israel and other countries [9,19,24]. The carrier rate is different in the different ethnic groups; this could be impacting the clinical course and manifestation of the disease. 

In the present study, we highlighted several important ethnicity-related factors that affect IBD patients. Understanding differences among ethnic groups and migrated people is important to help clinicians diagnose and treat patients. Special consideration of these populations is needed to decrease the prevalence of these diseases and prevent complications. Interventions should be carried out on several levels: First, an improvement in healthcare delivery for specific populations with high IBD incidence is required; second, implementation of personalized medicine, using electronic healthcare records, and improved access to medicine can be used to shorten the time from symptom onset to diagnosis and to improve treatment, assist in follow-up, and avoid disease complications [17]. 

The main strength of this study is its large number of patients and its population-based design. However, several limitations should be noted. First, this study had a retrospective design; second, due to the big data nature of this study, information regarding disease severity, as well as endoscopic and histopathologic data, are missing. Other limitations of the study are the lack of data regarding the Montreal classification and the bias related to missing variables not included in the database. 

## 5. Conclusions

Important disparities were found regarding IBD characteristics, comorbidities, treatment, complications, and outcomes among different ethnic groups. Arab IBD patients are younger in age at diagnosis, with male predominance, treated more frequently with azathioprine, and have a higher rate of anti-TNF treatment. 

## Figures and Tables

**Figure 1 jpm-13-01008-f001:**
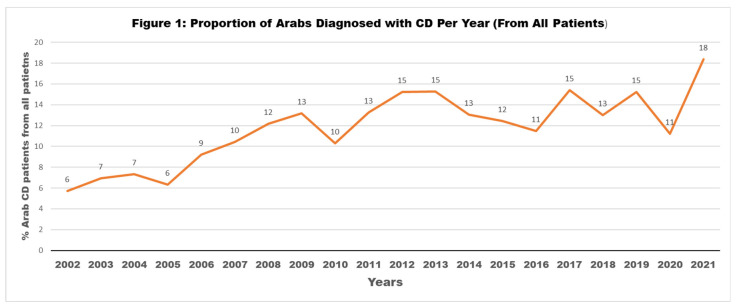
Proportion of Arabs diagnosed with CD per year (from all patients).

**Figure 2 jpm-13-01008-f002:**
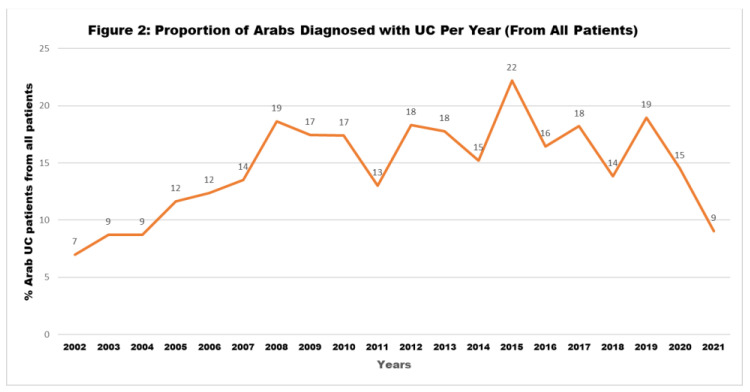
Proportion of Arabs diagnosed with UC per year (from all patients).

**Table 1 jpm-13-01008-t001:** Baseline characteristics of Arab and Jewish Crohn’s disease patients.

Crohn’s Disease Patients Characteristics	Arabs *n* = 1263 (%)	Jews *n* = 11,625 (%)	*p*-Value
Age, mean (range)	46.39 (84)	51.61 (99)	<0.001
Age at diagnosis, mean(range)	36.11 (78)	39.98 (79)	<0.001
Age group at diagnosis 0–10 10–17 18–65 >65	18 (1.4) 133 (10.5) 1026 (81.2) 86 (6.8)	83 (0.7) 1095 (9.4) 8839 (76) 1608 (13.8)	<0.001
Gender, male (%, 95% CI)	752 (59.5, 0.568–0.623)	5663 (48.7, 0.478–0.496)	<0.001
Smoking (past or current)	512 (40.5)	4831(41.6)	0.485
Appendectomy	16 (1.26)	107 (0.92)	0.529
BMI mean (range)	24.2 (48)	24 (48)	0.181
Numbers of admission to hospital, mean ± SD	5.96 ± 8.4	4.7 ± 6.9	<0.001
Hospitalized: any time	93 (7.4)	1549 (13.3)	<0.001
Uveitis	23 (1.8)	651 (5.6)	<0.001
Scleritis	13 (1)	194 (1.7)	0.086
Erythema nodosum	274 (2.4)	18 (1.4)	0.035
Pyoderma gangrenosum	3 (0.2)	22 (0.2)	0.711
Primary sclerosing cholangitis	18 (1.4)	177 (1.5)	0.788
Arthritis	184 (14.6)	1851 (15.9)	0.210
Perianal abscess	122 (9.7)	851 (7.3)	0.003
Anal fissure	78 (6.2)	572 (4.9)	0.053
Abdominal abscess	45 (3.6)	299 (2.6)	0.038
Death (%, 95% CI)	106 (8.4, 0.069–0.101)	1189 (10.2, 0.097–0.108)	0.039
Age at death, mean ± SD	64 ± 18.4	74.7 ± 15.2	<0.001

**Table 2 jpm-13-01008-t002:** Treatment of Crohn’s disease among Arab and Jewish patients.

Characteristics: Crohn’s Disease	Arabs *n* = 1263 (%)	Jews *n* = 11,625 (%)	*p*-Value
5-ASA	837 (66.3)	7535 (64.8)	0.304
Azathioprine	334 (26.4)	2382 (20.5)	<0.001
Mercaptopurine	167 (13.2)	2320 (20)	<0.001
Anti-TNF treatment	482 (38.2)	4698 (40.4)	0.121
Age at first anti-TNF	32.7 ± 13.6	36 ± 16.3	<0.001
Time from diagnosis to first anti-TNF treatment, mean ± SD	1121 ± 1334	1409 ± 1484	<0.001
Methotrexate	98 (7.8)	750 (6.5)	0.075
Budesonide	39 (3.1)	300 (2.6)	0.285
Prednisone	968 (76.6)	7451 (64.1)	<0.001
Prednisone: count, mean ± SD	10.24 ± 21.9	8.46 ± 20.8	0.004
Vedolizumab	91 (7.2)	1200 (10.3)	<0.001
Age at vedolizumab treatment, mean ± SD	39.18 ± 14.5	44.9 ± 18	0.001
Time from diagnosis to vedolizumab treatment, mean ± SD	2451 ± 1664	2700 ± 1867	0.430
Ustekinumab	76 (6)	966 (8.3)	0.005
Age at ustekinumab treatment mean ± SD	35.3 ± 12.9	42.1 ± 17	<0.001
Time from diagnosis to ustekinumab treatment, mean ± SD	2780 ± 1713	3191 ± 1878	0.065
Any bowel surgery (%, 95% CI)	62 (4.9, 0.038–0.062)	623 (5.4, 0.50–0.058)	0.498

**Table 3 jpm-13-01008-t003:** Laboratory values at the diagnosis time of Crohn’s disease.

	Arabs *n* = 1263	Jews *n* = 11,625	*p*-Value
CRP	10.34 ± 31.1	5.67 ± 18.3	<0.001
Calprotectin *	560.32 ± 1069	599.87 ± 1565	0.680
HB	12.76 ± 2	12.87 ± 1.7	0.084
WBC	8.99 ± 3.4	8.53 ± 3.3	<0.001
PLT	326.2 ± 121	316.56 ± 11.6	0.007
AST	21.78 ± 27.3	20.8 ± 30	0.296
ALT	21.89 ± 23	20.14 ± 43.2	0.157
Albumin	3.93 ± 0.56	4 ± 0.5	<0.001
Vitamin B12	322.22 ± 184	332 ± 175	0.104
Iron	50.73 ± 34	60.58 ± 38	<0.001
Ferritin	89.1 ± 123	84.34 ± 167	0.347

All values are mean ± SD. * Calprotectin values were available among 27% of the CD patients. CRP = C-reactive protein; WBC = white blood cells count; PLT = platelets; AST = aspartate transaminase; ALT = alanine aminotransferase.

**Table 4 jpm-13-01008-t004:** Baseline characteristics of Arab and Jewish ulcerative colitis patients.

Ulcerative Colitis Patients Characteristics	Arabs *n* = 1461 (%)	Jews 10,920 (%)	*p*-Value
Age, mean (range)	48.8 (85)	58 (97)	<0.001
Age at diagnosis, mean (range)	37.9 (78)	45.6 (80)	<0.001
Age group at diagnosis 0–10 10–17 18–65 >65	8 (0.5) 94 (6.4) 1251 (85.6) 108 (7.4)	41 (0.4) 607 (5.6) 8133 (74.5) 2139 (19.6)	<0.001
Gender, male (%, 95% CI)	769 (52.6, 0.500–0.552)	5256 (48.1, 0.472–0.491)	0.001
Smoking (past or current)	406 (27.8)	3879 (35.5)	<0.001
Appendectomy	15 (1)	80 (0.7)	0.237
BMI, mean (range)	25.2 (48)	24.8 (49)	0.017
Numbers of admission to hospital, mean ± SD	4 ± 6.2	3.87 ± 6.1	0.350
Hospitalized: any time	65 (4.4)	1085 (9.9)	<0.001
Uveitis	40 (2.4)	779 (7.1)	<0.001
Scleritis	13 (0.9)	122 (1.1)	0.432
Erythema nodosum	20 (1.4)	137 (1.3)	0.714
Pyoderma gangrenosum	7 (0.5)	29 (0.3)	0.155
Primary sclerosing cholangitis	24 (1.6)	199 (1.8)	0.628
Arthritis	216 (14.8)	2139 (19.6)	<0.001
Perianal abscess	54 (3.7)	310 (2.8)	0.68
Anal fissure	39 (2.7)	226 (2.1)	0.137
Abdominal abscess	12 (0.8)	81 (0.7)	0.741
Death (%, 95% CI)	121 (8.3, 0.069–0.098)	1548 (14.2, 0.135–0.148)	<0.001
Age at death, mean ± SD	66.67 ± 17.7	78.28 ± 12.9	<0.001

**Table 5 jpm-13-01008-t005:** Treatments of UC patients among Arab and Jewish patients.

Characteristics	Arabs *n* = 1461 (%)	Jews 10,920 (%)	*p*-Value
5-ASA	1324 (90.6)	9988 (91.5)	0.282
Azathioprine	259 (17.7)	1101 (10.1)	<0.001
Mercaptamine	159 (10.9)	1122 (10.3)	0.472
Anti-TNF treatment	261 (17.9)	1490 (13.6)	<0.001
Age at anti-TNF beginning, mean ± SD	35.4 ± 13.8	40.3 ± 18	<0.001
Time diagnosis to anti-TNF treatment, days, mean ± SD	1643 ± 1497	1887 ± 1670	0.018
Methotrexate	53 (3.6)	371 (3.4)	0.650
Budesonide	37 (2.5)	244 (2.2)	0.471
Prednisone	1030 (70.5)	6016 (55.1)	<0.001
Vedolizumab	152 (10.4)	978 (9)	0.071
Age at vedolizumab beginning	35.5 ± 14.7	44.8 ± 18.5	<0.001
Time diagnosis to vedolizumab TNF treatment, days, mean ± SD	2019 ± 1672	2761 ± 2063	<0.001
Any surgery (%, 95% CI)	30 (2.1, 0.014–0.029))	172 (1.6, 0.013–0.018)	0.175
Total colectomy	14 (1)	60 (0.5)	0.057

**Table 6 jpm-13-01008-t006:** Laboratories values at the time of UC diagnosis.

	Arabs *n* = 1461	Jews *n* = 10,920	*p*-Value
CRP	5.75 ± 19.2	4 ± 17.5	0.002
Calprotectin *	1073.6 ± 3246	743.16 ± 1946	0.104
HB	12.87 ± 2	13.1 ± 1.75	<0.001
WBC	8.28 ± 3	7.9 ± 3.3	<0.001
PLT	299.1 ± 107	282.8 ± 96	<0.001
AST	21.75 ± 26.8	21.76 ± 20.7	0.995
ALT	22.24 ± 37	21.14 ± 26.1	0.275
Albumin	4.12 ± 0.58	4.14 ± 0.51	0.066
Vitamin B12	338.1 ± 198	345.1 ± 178	0.265
Iron	57.79 ± 38.1	71.62 ± 38.8	<0.001
Ferritin	69.68 ± 135	74.10 ± 126	0.255

All values are mean ± SD. * Calprotectin values were available among 22% of the UC patients. CRP = C-reactive protein; WBC = white blood cells count; PLT = platelets; AST = aspartate transaminase; ALT = alanine aminotransferase.

**Table 7 jpm-13-01008-t007:** Uni- and multivariable analysis of mortality and hospitalization among Crohn’s disease and ulcerative colitis patients.

	Univariate Analysis			Multivariate Analysis		
Mortality for Crohn’s Disease	OR	95% CI	*p*-Value	OR	95% CI	*p*-Value
Arab Ethnicity *	0.804	0.653–0.990	0.40	0.810	0.657–0.997	0.047
Gender #	0.958	0.854–1.074	0.461	1.051	0.936–1.181	0.401
Smoking	1.108	0.987–1.244	0.803	1.114	0.991–1.253	0.069
Hospitalization for Crohn’s Disease						
Ethnicity *	0.517	0.416–0.643	<0.001	0.524	0.421–0.652	<0.001
Gender #	1.058	0.954–1.174	0.283	1.085	0.977–1.205	0.128
Smoking	1.426	1.285–1.582	<0.001	1.439	1.296–1.598	<0.001
Mortality for Ulcerative Colitis						
Ethnicity *	0.547	0.450–0.664	<0.001	0.544	0.448–0.661	<0.001
Gender #	0.808	0.729–0.986	<0.001	0.811	0.730–0.901	<0.001
Smoking	1.147	1.030–1.276	0.012	1.084	0.972–1.208	0.149
Hospitalization for Ulcerative Colitis						
Ethnicity *	0.422	0.327–0.545	<0.001	0.429	0.332–0.554	<0.001
Gender #	1.042	0.922–1.176	0.510	1.058	0.935–1.197	0.372
Smoking	1.190	1.050–1.349	0.006	1.176	1.035–1.336	0.013

* Jewish patients as reference, # males as reference.

## Data Availability

No additional data are available.

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
