# Peer review of "Ethnic Issues and Disparities in Inflammatory Bowel Diseases: What Can We Learn from the Arab Population in Israel?"

_jpm, 2023, doi:10.3390/jpm13061008_

Round 1

Reviewer 1 Report

Thank you for your interesting article 

some comments:

First line of introduction:  I do not think the role ethnicity is an emerging field in gastroenterology. This is well recognised for many years. Need to reference more than just your data on this and include more international data eg US and UK data and data on NODS and ethnicity

 Aniwan S, Harmsen WS, Tremaine WJ, Loftus EV Jr. Incidence of inflammatory bowel disease by race and ethnicity in a population-based inception cohort from 1970 through 2010. Therap Adv Gastroenterol. 2019 Feb 6;12:1756284819827692. doi: 10.1177/1756284819827692. PMID: 30792818; PMCID: PMC6376543.

Misra R, Limdi J, Cooney R, Sakuma S, Brookes M, Fogden E, Pattni S, Sharma N, Iqbal T, Munkholm P, Burisch J, Arebi N. Ethnic differences in inflammatory bowel disease: Results from the United Kingdom inception cohort epidemiology study. World J Gastroenterol. 2019 Oct 28;25(40):6145-6157. doi: 10.3748/wjg.v25.i40.6145. PMID: 31686769; PMCID: PMC6824277.

Karban A, Waterman M, Panhuysen CI, Pollak RD, Nesher S, Datta L, Weiss B, Suissa A, Shamir R, Brant SR, Eliakim R. NOD2/CARD15 genotype and phenotype differences between Ashkenazi and Sephardic Jews with Crohn's disease. Am J Gastroenterol. 2004 Jun;99(6):1134-40. doi: 10.1111/j.1572-0241.2004.04156.x. PMID: 15180737.

Intro line 40 - references needed 

Last sentence of first paragraph in intro needs reference 

Suggest change to 'change in diet' rather than 'change in nutrition' line 49 page 2.  Do you have any reference for all these statements re changes to arab population?

Line 56

Please change- Increase in incident ( but not amongst those diagnosed). Please state incidence as per referenced paper.

need to add p values for line 108 ie rates of anal fissures etc 

Line 270 - you mention genetic treatments - there are none- please reword.

In tables - please label appropriate column with %

I do not think SD are suitable for a lot of your results - need mean/range eg for age and BMI rather than SD.  Please consider changing.

No information is given regarding proportion of patient in which information is available eg was f.cal available on all patients at diagnosis ?

If information not available on all patients - please add this to limitations

You mention bloods used were the bloods taken nearest in time to diagnosis - can you give a range of timing ie were most within a week of diagnosis ?

Do you disease classification available eg Montreal? 

If not please add to limitations 

Genetic data would also enhance this study- do you have information on different Jewish groups given the known relationship in CD eg Ashkenazi vs Sephardic Jewish population

Could there be any other reason for difference between M:F incidence?

Please compare this to worldwide data 

Finally the referencing is inconsistent - please make sure all references are done in the same style 

Rewrite sentence 225-226 - poor wording

Line 276-281- needs to be rewritten- duplication of some parts of sentence

Author Response

Reviewer 1:

First line of introduction:  I do not think the role ethnicity is an emerging field in gastroenterology. This is well recognised for many years. Need to reference more than just your data on this and include more international data eg US and UK data and data on NODS and ethnicity

 Aniwan S, Harmsen WS, Tremaine WJ, Loftus EV Jr. Incidence of inflammatory bowel disease by race and ethnicity in a population-based inception cohort from 1970 through 2010. Therap Adv Gastroenterol. 2019 Feb 6;12:1756284819827692. doi: 10.1177/1756284819827692. PMID: 30792818; PMCID: PMC6376543.

Misra R, Limdi J, Cooney R, Sakuma S, Brookes M, Fogden E, Pattni S, Sharma N, Iqbal T, Munkholm P, Burisch J, Arebi N. Ethnic differences in inflammatory bowel disease: Results from the United Kingdom inception cohort epidemiology study. World J Gastroenterol. 2019 Oct 28;25(40):6145-6157. doi: 10.3748/wjg.v25.i40.6145. PMID: 31686769; PMCID: PMC6824277.

Karban A, Waterman M, Panhuysen CI, Pollak RD, Nesher S, Datta L, Weiss B, Suissa A, Shamir R, Brant SR, Eliakim R. NOD2/CARD15 genotype and phenotype differences between Ashkenazi and Sephardic Jews with Crohn's disease. Am J Gastroenterol. 2004 Jun;99(6):1134-40. doi: 10.1111/j.1572-0241.2004.04156.x. PMID: 15180737.

Response:

 We like to thank the reviewer for this comment. We agree it is a well recognized field and we wanted to emphasize the importance of this ethnicity in field of gastroenterology. As you suggested we have included additional studies of international data.

The sentence was rewritten as: (line 37)

Race and ethnicity play important roles in the epidemiology and outcomes of diseases, this issue has been investigated in different diseases in the field of gastroenterology.

In addition, the suggested references were cited, and additional national data were included.

Line 42

In addition, significant racial and ethnic differences of IBD were found over four decades (between 1970 and 2010) in the United States population, with an increase in the incidence rate by 39% among whites and 139% among nonwhites [11]. Another study found a high age-adjusted difference in incidence of IBD among Indians compared to White Europeans and Pakistanis in the United Kingdom [12].

Line 51

However, disparities regarding other aspects such as NOD2/CORD15 and serological markers (ASCA/ANCA) were found among different ethnic groups in Israel and other countries [8-10,13,14,18].

Intro line 40 - references needed 

Several references are included in the line 40, the sentence is general and included the refences 1-10, but the same references are cited later in the manuscript.

Last sentence of first paragraph in intro needs reference 

Added

Suggest change to 'change in diet' rather than 'change in nutrition' line 49 page 2.  Do you have any reference for all these statements re changes to arab population?

 Corrected to change in diet,

We added to references regarding health (diseases) of the Arab population.

Line 56

Please change- Increase in incident ( but not amongst those diagnosed). Please state incidence as per referenced paper.

Were changed and the incidences were added according to the referenced paper.

need to add p values for line 108 ie rates of anal fissures etc 

 added

Line 270 - you mention genetic treatments - there are none- please reword.

Thank you for the note, the end of the sentence doesn’t make sense and we deleted “including genetic, environmental, and health based

 In tables - please label appropriate column with %

Added

I do not think SD are suitable for a lot of your results - need mean/range eg for age and BMI rather than SD.  Please consider changing.

For age and BMI, we changed to “range”.

No information is given regarding proportion of patient in which information is available eg was f.cal available on all patients at diagnosis ?

If information not available on all patients - please add this to limitations

You mention bloods used were the bloods taken nearest in time to diagnosis - can you give a range of timing ie were most within a week of diagnosis ?

We would like to thank the reviewer for the important comments.

Regarding the labs, we included labs values if it was available in at least 80% of the patients, this met for all labs but not for Calprotectin, because calprotectin was not available for the entire study period. The labs included within six months before or after the diagnosis.

We add the following in the methods section:

nearest to the time of IBD diagnosis within six months before or after diagnosis, was recorded, the labs value for every variable included in the tables if it was available among at least of 80% of the patients, except Calprotectin which was not available in the entire study period.

We add the following notes in the tables:

* Calprotectin values were available among 27% of the CD patients

* Calprotectin values were available among 22% of the UC patients.

Do you disease classification available eg Montreal? 

If not please add to limitations 

Unfortunately, no data regarding Montreal classification was available due to the design and nature of the data. As suggested, we added it as a limitation of the study.

Genetic data would also enhance this study- do you have information on different Jewish groups given the known relationship in CD eg Ashkenazi vs Sephardic Jewish population.

We add the following paragraph, to stay in the same line of the discussion we cited the genetic information in different ethnic groups.

Different NOD2/CARD15 carrier rates were found among different ethnic groups, and different carrier rates were also found among subgroups of the Jewish population. A NOD2/CARD15 carrier rate of 32% among CD Bedouin patients was reported, 47.4% among Ashkenazi Jewish CD patients, 27.4% among Sephardic Jews [7,12], and 7-16% among Caucasian CD patients [13]. A lower carrier rate of 8.2% was found among Arabs in other regions in Israel and other countries [9,19, 25]. The carrier rate is different in the different ethnic groups, this could be impacting the clinical course and manifestation of the disease.

Could there be any other reason for difference between M:F incidence?

Please compare this to worldwide data 

Among Crohn’s disease patient in our study a higher proportion of males was found, in a previous study from Zvidi et al and other studies, a same finding was reported.

We add the following paragraph in the discussion section

More Arab CD and UC patients were males than Jewish patients. Another study investigated the epidemiology of IBD among Arab patients in Israel and reported a higher rate of males among CD patients but not among UC patients [2], data from Western and Eastern Europe a higher rate of males was found among newly diagnosed UC patients but a males predominance was found among CD patients only in Eastern patients [1], in the meta-analysis of studies from the Arab World a male predominance reported, with 41-60% of the IBD patients [6] and other studies showed different findings according to the ethnic groups [10,11].  The higher frequency rate of males is may be explained by different factors; exposure rates to specific risk factors such as smoking or specific foods; however, it remains to be determined in future studies.

Finally the referencing is inconsistent - please make sure all references are done in the same style 

Corrected

Comments on the Quality of English Language

Rewrite sentence 225-226 - poor wording

The sentence was simplified as:

However, in our study, the percentage of Arab CD patients increased continuously in the last two decades.

Line 276-281- needs to be rewritten- duplication of some parts of sentence

The sentence rewritten as:

second, implementation of personalized medicine, using electronic health care records, and improvement in access to healthcare can be used to shorten the time from symptom onset to diagnosis and to improve treatment, follow-up and, avoiding disease complications.

Reviewer 2 Report

GENERAL COMMENTS

The grammar of the entire manuscript should be reviewed.

Please review all references in the text, as some of them are not in brackets.

TITLE

OK.

ABSTRACT

It is stated “We aimed to compare the clinical characteristics, complications, and outcomes among different ethnic groups sharing the same healthcare system”. However, only Arab vs. Jewish patients were compared, so I suggest to change the Aims to more accurately reflect this particular comparison.

This same comment is applicable to the Conclusions section of the abstract.

Please use the term mercaptopurine (without 6-).

INTRODUCTION

The authors point out that “We aimed to investigate IBD among the Arab population in Israel in 60 terms of epidemiology, comorbidities, extraintestinal manifestations, complications, treatments, and all-cause mortality”. Please see my previous comment in the Abstract section and change accordingly the wording of the objectives (Arab vs. Jewish patients, in particular).

METHODS

More detailed information should be included (regarding the database, the way the patients are selected/included, the way variables are included, etc.). This is important to correctly assess the accuracy/reliability of the data.

Some information on the inclusion and exclusion criteria should be provided.

Statistical analysis:

-       The 95% confidence interval should be calculated for the most relevant percentages.

-       The following information does not belong to the Statistical analysis section: “The study was carried out 88 in accordance with the principles of the Helsinki Declaration. The study protocol was ap-89 proved by the Institutional Helsinki Committee, approval number 97-21. Informed con-90 sent was waived due to the retrospective design of the study”.

-       I suggest performing a multivariate (logistic regression) analysis to more accurately evaluate the role of the ethnic groups (Arab vs. Jewish, which should be included as independent variables in the model) on some relevant outcomes (such as for example hospitalization of mortality, which should be included as dependent variable).

RESULTS

Were patients with an indeterminate/unclassifiable colitis excluded from this study?

DISCUSSION

Some paragraphs are too long and should be divided into shorter ones.

In the section dealing with limitations, I would highlight the possible biases that could arise due to the retrospective nature of the study (including for example different variables that were not included in the database).

REFERENCES

OK.

TABLES

OK.

FIGURES

Figure 2 (Proportion of Arab diagnosed with UC per year): the low numbers in 2020 and, especially, in 2021 suggest that the inclusion of patients in the database was not as complete as in previous years, which could create a bias. Please confirm. The hypothesis involving the COVID-19 pandemic seems not to be the only one.

.

Author Response

The response letter to Reviewer 2 is attached. 

Round 2

Reviewer 2 Report

.